# Analysis of the Effect of Pulse Length and Magnetic Field Strength on Nonlinear Thomson Scattering

Haokai Wang [1] 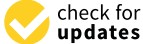, Feiyang Gu [1], Yi Zhang [2], Yubo Wang [1], Qingyu Yang [1] and Youwei Tian [3,*]

[1] Bell Honors School, Nanjing University of Posts and Telecommunications, Nanjing 210023, China; q21010206@njupt.edu.cn (H.W.); q21010204@njupt.edu.cn (F.G.); b20060511@njupt.edu.cn (Y.W.); b20021922@njupt.edu.cn (Q.Y.)

[2] School of Communications and Information Engineering, Nanjing University of Posts and Telecommunications, Nanjing 210023, China; q21010211@njupt.edu.cn

[3] College of Science, Nanjing University of Posts and Telecommunications, Nanjing 210023, China

[*] Correspondence: tianyw@njupt.edu.cn

**Abstract:** In this paper, two parameters of the applied magnetic field—magnetic field strength and pulse length—are modified, the spatial properties of electron trajectories and radiation are studied, and conclusions are drawn. Under the premise that the radius of the laser pulse waist $b_0 = 4\lambda_0$, and that the peak amplitude $a_0 = 5$ corresponds to the peak laser intensity $I_L = 3.45 \times 10^{19}$ W/cm$^2$ of the electrons, obtaining high-energy and highly collimated X-rays can be realized by increasing the pulse length up to $6\lambda_0$ and increasing the magnitude of the applied magnetic field, or by continuously increasing the pulse length and applying a smaller magnetic field.

**Keywords:** laser physics; circular polarization; tightly focused laser; applied magnetic field

## 1. Introduction

Since the first laser was introduced in 1960, research on ultra-short and ultra-intense laser pulses has been pursued [1,2]. With the advent of self-mode-locked technology [3] and chirped pulse amplification [4], there has been a tremendous breakthrough in laser physics. As the latest frontier of international laser science, ultra-short laser pulses are currently the focus of attention due to their significant potential in the study of ultra-fast processes. It is now possible to generate attosecond laser pulses with laser field intensities of up to $10^{15}$ W/cm$^2$, advancing the physics of ultra-fast lasers [5]. In the important field of intense laser–matter interaction, relativistic nonlinear Thomson scattering has attracted many scholars as an important means of modulating X-rays and $\gamma$-rays. A very wide range of applications have been achieved in the fields of biomedicine [6], atomic physics [7], modern astrophysics [8,9], nuclear physics [10,11], laser detection [12,13], and science and industry [14].

Nonlinear Thomson scattering is a scattering process applied to relativistic charged particles in a strong electromagnetic field, where the nonlinearity is significant due to the Doppler effect. In past work, the effect of various parameters of the laser pulse on the electron radiation has been studied. Chen et al. [15] investigated the effect of laser intensity on the space radiation of high-energy electrons. Wang et al. [16] and Yu et al. [17] investigated the effects of pulse length and girdle radius, respectively. Zhang et al. [18] and Yan et al. [19] investigated the effects of pulse length and the initial position of electrons on the peak radiated power and spatial properties of nonlinear Thomson scattering, respectively. In addition, K.P. Singh [20] and Gupta et al. [21] investigated the introduction of a static magnetic field and a super-intense pulsed magnetic field on top of a laser field, respectively. In contrast, Zhang et al. [22] introduced an applied variable magnetic field and investigated the effect of beam waist radius on electron radiation characteristics based

on a previous study introducing an ultra-intense laser-pulsed magnetic field and a static magnetic field in the laser field.

However, in previous work, on the one hand, the authors studied the effect of this parameter on the nonlinear Thomson scattering only from a certain point of view, i.e., the electron radiation, without discovering the intrinsic connection between the individual radiation characteristics; on the other hand, they only modulated the parameters of the laser, i.e., without changing the applied magnetic field. Therefore, it is reasonable that the collimation of their modulated electron radiation is poor, and the peak power is not high enough. In this way, the idea of combining the analysis of the spatial radiation with the motion trajectories of nonlinear Thomson scattering becomes particularly important; through this method, we can not only study the radiated spatial radiation itself but also the process of its radiation. Previous studies have proven that the introduction of an applied magnetic field can substantially improve the nonlinear Thomson scattering radiation characteristics. On this basis, this paper mainly studies and analyzes the relationship between the electron motion trajectory and radiation generation, as well as the spatial radiation of the electron itself, by changing the strength and pulse length of the applied magnetic field. This paper explains, for the first time, the correspondence between the magnitude of electron radiation at each moment and the trajectory of the electron, thus explaining the process of electron radiation. Figure 1 shows the schematic diagram of this paper.

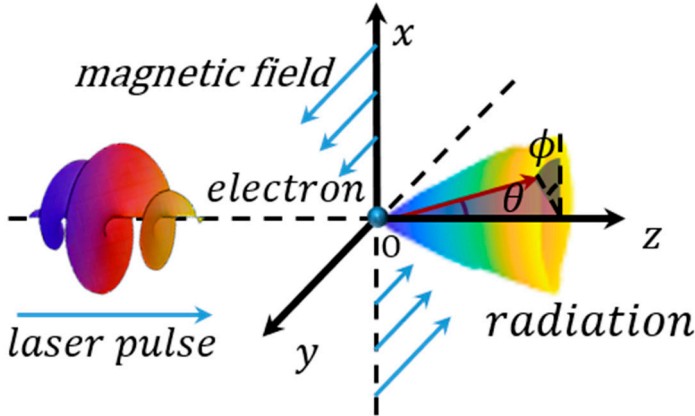

**Figure 1.** Schematic diagram of nonlinear Thomson scattering.

In summary, this paper focuses on the motion trajectory and spatial radiation of electrons from a unique perspective, investigating, for the first time, the integrated effect of pulse length and the applied variable magnetic field on electron-excited radiation. On this basis, we select the optimal parameter combination of the composite field of the laser field and the magnetic field, thus helping us to guide the modulation of high-peak-power, high-collimation rays.

## 2. Materials and Methods

First, it should be noted that all the definitions of the following formulae are normalized separately with $k_0^{-1}$ for spatial coordinates and $\omega_0^{-1}$ for time coordinates, where $\omega_0 = 2\pi c / \lambda_0$ corresponds to the laser frequency, $k_0 = 2\pi / \lambda_0$ corresponds to the laser wavenumber, and $\lambda_0$ is the laser wavelength, specifically 1 μm, with $c$ being the speed of light.

The normalized vector potential of a focused Gaussian pulsed laser electric field with an applied magnetic field can be expressed as follows:

$$\vec{a} = a_0 exp\left(-\frac{\eta^2}{L^2} - \frac{\rho^2}{b_0{}^2\left(1 + \frac{z^2}{Z_f{}^2}\right)}\right)\frac{1}{\sqrt{1 + \frac{z^2}{Z_f{}^2,}}}\left[cos(\varphi)\vec{x} + \delta sin(\varphi)\vec{y}\right] + B_0 x\vec{y} \quad (1)$$

where $a_0$ is the normalized laser amplitude by $m\frac{c^2}{e}$; $m$ and $e$ denote the electron rest mass and net charge; $L$ and $b$ denote the pulse length and beam waist radius of the laser; and phase $\varphi$ can be expressed as follows: $\varphi = \eta + \varphi_0 - \theta + \frac{\rho^2}{2} \times R(z)$. $\eta = z - t$. $\rho^2 = x^2 + y^2$. $\theta = arctan\frac{z}{Z_f}$. $Z_f = \frac{b_0{}^2}{z}$. $R(z) = \frac{z}{Z_f{}^2 + z^2}$. $b = b_0\sqrt{1 + \frac{z^2}{Z_f{}^2}}$.

We set

$$a_L = a_0\frac{b_0}{b}\exp\left(-\frac{\eta^2}{L^2} - \frac{\rho^2}{b^2}\right); \quad (2)$$

then, there is

$$\begin{cases} a_x = a_L cos\varphi \\ a_y = \delta a_L sin(\varphi) + B_0 x \end{cases}. \quad (3)$$

Relativistic electron motion in a strong laser field is described by the Lorentz equation:

$$d_t p = \partial_t p + (u\cdot\nabla)p = -E - u \times B, \quad (4)$$

where $p = \gamma u$ is the electron momentum normalized by $mc$; $u$ is the electron velocity normalized by the speed of light $c$; $\gamma$ is the relativistic factor of the electron; and the time and space coordinates are normalized by $\omega^{-1}$ and $k^{-1}$ (not explained below).

Combined with the above equations, the phase and vector potentials can be partially differentiated as follows:

$$\begin{cases} \partial_t\varphi = -1 \\ \partial_x\varphi = xR(z) \\ \partial_y\varphi = yR(z) \\ \partial_z\varphi = 1 - \frac{Z_f}{Z_f{}^2 + z^2} - \frac{\rho^2\left(z^2 - Z_f{}^2\right)}{Z_f{}^2 + z^2,} \end{cases} \quad (5)$$

$$\begin{cases} \partial_t a_L = \frac{2\eta}{L^2}a_L \\ \partial_x a_L = \frac{-2x}{b^2}a_L \\ \partial_y a_L = \frac{-2y}{b^2}a_L \\ \partial_z a_L = a_L\left(-\frac{\partial_z b}{b} + \frac{2(x^2 + b^2)\partial_z b}{b^3} - \frac{2\eta}{L^2}\right), \end{cases} \quad (6)$$

$$\begin{cases} \partial_t a_x = \partial_t a_L cos\varphi - \partial_t\varphi a_L sin\varphi \\ \partial_y a_x = \partial_y a_L cos\varphi - \partial_y\varphi a_L sin\varphi \\ \partial_z a_x = \partial_z a_L cos\varphi - \partial_z\varphi a_L sin\varphi \\ \partial_t a_y = -\delta[\partial_t a_L sin\varphi + \partial_t\varphi a_L cos\varphi] \\ \partial_x a_y = -\delta(\partial_x a_L sin\varphi + \partial_x\varphi a_L cos\varphi) + B_0 \\ \partial_z a_y = -\delta[\partial_z a_L sin\varphi + \partial_z\varphi\ a_L cos\varphi] \\ \partial_t a_z = -\left\{\frac{2x}{bb_0}\partial_t a_L sin(\varphi + \theta) + \frac{2x}{bb_0}a_L cos(\varphi + \theta)\partial_t\varphi + \delta\left[\frac{2y}{bb_0}\partial_t a_L cos(\varphi + \theta) - \frac{2y}{bb_0}a_L sin(\varphi + \theta)\partial_t\varphi\right]\right\} \\ \partial_x a_z = -\left\{\frac{2}{bb_0}a_L sin(\varphi + \theta) + \frac{2x}{bb_0}\partial_z a_L sin(\varphi + \theta) + \frac{2x}{bb_0}a_L cos(\varphi + \theta)\partial_x\varphi + \delta\left[\frac{2y}{bb_0}\partial_x a_L cos(\varphi + \theta) - \frac{2y}{bb_0}a_L sin(\varphi + \theta)\partial_x\varphi\right]\right\} \\ \partial_y a_z = -\left\{\frac{2x}{bb_0}\partial_y a_L sin(\varphi + \theta) + \frac{2x}{bb_0}a_L cos(\varphi + \theta)\partial_y\varphi + \delta\left[\frac{2}{bb_0}a_L cos(\varphi + \theta) + \frac{2y}{bb_0}\partial_y a_L cos(\varphi + \theta) - \frac{2y}{bb_0}a_L sin(\varphi + \theta)\partial_y\varphi\right]\right\} \end{cases} \quad (7)$$

Combining Equations (2)–(7) and the Coulomb norm condition $\nabla\cdot a = 0$, the decomposition is unfolded in a three-dimensional spatial coordinate system, and a set of differential equations is used to realize the spatial and temporal decomposition of the electron motion as follows:

$$\begin{cases} \gamma d_t u_x = \left(1 - u_x^2\right)\partial_t a_x + u_y\left(\partial_y a_x - \partial_x a_y\right) + u_z(\partial_z a_x - \partial_x a_z) - u_x u_y \partial_t a_y - u_x u_z \partial_t a_z \\ \gamma d_t u_y = \left(1 - u_y^2\right)\partial_t a_y + u_x\left(\partial_x a_y - \partial_y a_x\right) + u_z\left(\partial_z a_y - \partial_y a_z\right) - u_x u_y \partial_t a_x - u_y u_z \partial_t a_z \\ \gamma d_t u_z = \left(1 - u_z^2\right)\partial_t a_z + u_x(\partial_x a_z - \partial_z a_x) + u_y\left(\partial_y a_z - \partial_z a_y\right) - u_x u_z \partial_t a_x - u_y u_z \partial_t a_y \\ \qquad\qquad d_t\gamma = u_x \partial_t a_x + u_y \partial_t a_y + u_z \partial_t a_z. \end{cases} \quad (8)$$

An electron undergoing relativistic accelerated motion emits radiation, which has a power per-unit solid angle formula:

$$\frac{dP(t)}{d\Omega} = \left[\frac{\left|\vec{n} \times \left[\left(\vec{n} - u\right) \times d_t u\right]\right|^2}{\left(1 - \vec{n}\cdot u\right)^6}\right]_{t'} \quad (9)$$

$$\frac{dP(t)}{d\Omega} \propto \gamma^8 d_t u_x^2, \quad (10)$$

where the radiated power is normalized by $e^2\omega_0^2/4\pi c$; $\Omega$ is the unit solid angle; and the direction of radiation $\vec{n} = \sin(\theta)\cos(\varphi)\cdot\vec{x} + \sin(\theta)\sin(\varphi)\cdot\vec{y} + \cos(\theta)\cdot\vec{z}$.

The derivation of Equation (10) is given by [23].

$t'$ is the time at which the electron interacts with the laser pulse; the following relationship exists between $t'$ and t:

$$t = t' + R_0 - \vec{n}\cdot\vec{r}, \quad (11)$$

where $R_0$ is the distance of the observation point from the electron and $\vec{r}$ is the electron site vector. It is assumed that the observation point is far away from the point of interaction between the laser and the electron.

## 3. Results

In this section, we will discuss the effect of the laser pulse on the electron trajectories via the various parameters of an incident, tightly focused laser pulse in the presence of an applied magnetic field.

Our numerical simulation parameters are peak laser amplitude $a_0 = 5$ and beam waist radius $b_0 = 4\lambda_0$. $\lambda_0$ denotes the initial acting wavelength of the laser pulse, where $\lambda_0 = 1$ μm. The unit of the magnetic field $B_0$ is KT. We calculate the electron motion trajectories under different pulse lengths L and different magnetic field strengths $B_0$ by writing MATLAB R2020a programs based on this parameter.

### 3.1. Electron Motion and Corresponding Radiation Properties 1

Figure 2 shows the trajectory of electrons under the combined effect of laser pulse and magnetic field. The maximal coordinates of the corresponding *x*-axis and *y*-axis in the plots are both $\pm 10\lambda_0$.

The color of the line in Figure 2 indicates the magnitude of the radiation at each point on the trajectory, while the direction of the radiation is a section of the cone in the direction of the electron velocity. The blue line in the background of each figure indicates the magnitude of the relativistic factor $\gamma$ of the electron at the corresponding *z*-coordinate position. Please note that the magnitude of the relativistic factor $\gamma$ is here normalized by the maximum relativistic factor that is under the corresponding parameter. Therefore, in this study, we can only compare the relative magnitude under the same parameter, not the absolute magnitude in different plots, i.e., the magnitude under different parameters.

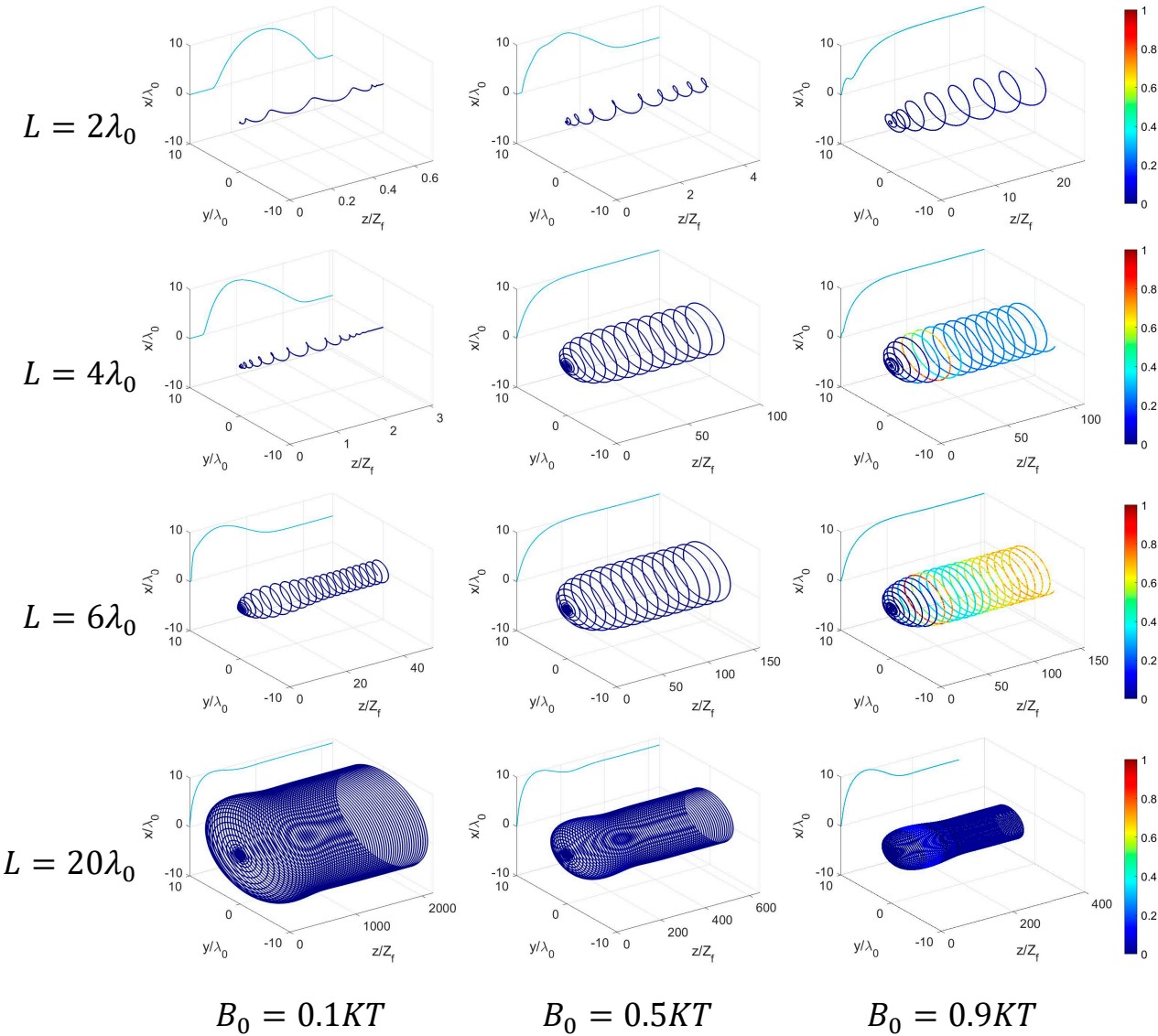

**Figure 2.** Trajectories of electrons and coincidence plots normalized to the maximum radiated power of all data.

Observing the differences between columns, we begin by analyzing the plots where pulse length $L \leq 6\lambda_0$ in Figure 2. As the magnetic field strength $B_0$ increases, the maximum energy radiated by the electron gradually increases, and the ratio of the magnitude of the relativistic factor $\gamma$ at the maximum radiation to the magnitude of the relativistic factor $\gamma$ after smoothing becomes smaller; while after pulse length $L > 6\lambda_0$, the ratio of the magnitude of the relativistic factor $\gamma$ at the maximum radiation to the magnitude of the relativistic factor $\gamma$ after smoothing becomes larger. From the trajectory diagram, we also find that the electron trajectory changes when $L > 6\lambda_0$. The radial radius of the electron begins to contract after reaching its maximum, corresponding to the trend of the relativistic factor $\gamma$.

Observing from the difference between rows, we can also find that as the pulse length L increases, the electron trajectory changes from the original olive shape to a regular cylindrical shape. The ratio of the magnitude of the relativistic factor $\gamma$ at the maximum radiation to the magnitude of the relativistic factor $\gamma$ after smoothing becomes larger; while for the portion of the pulse length $L > 6\lambda_0$, the radiation becomes smaller relative to $L = 6\lambda_0$. This indicates that there is an optimal pulse length parameter $L = 6\lambda_0$ for the tightly focused laser pulse under an applied magnetic field.

We also find that when the pulse length $L > 6\lambda_0$, the trend of the radial radius is different from the previous one. At pulse length $L \leq 6\lambda_0$, the maximum radial radius increases with the increasing magnetic field strength $B_0$ until the maximum radial radius equals the final radial radius. In contrast, for pulse lengths $L > 6\lambda_0$, the maximum radial radius decreases with the increasing magnetic field strength $B_0$.

### 3.2. Electron Motion and Corresponding Radiation Properties 2

The parameters in Figure 3 are the same as those in Figure 2. The difference is that the radiated power in Figure 2 is normalized by the maximum radiated power of all parameters, while the radiated power in Figure 3 is normalized by the maximum radiated power of itself. In this way, from Figure 3, the magnitude of the radiated power and its difference between the front end and the back end of the motion can be observed and compared. This enables us to study the causes of radiation generation more successfully.

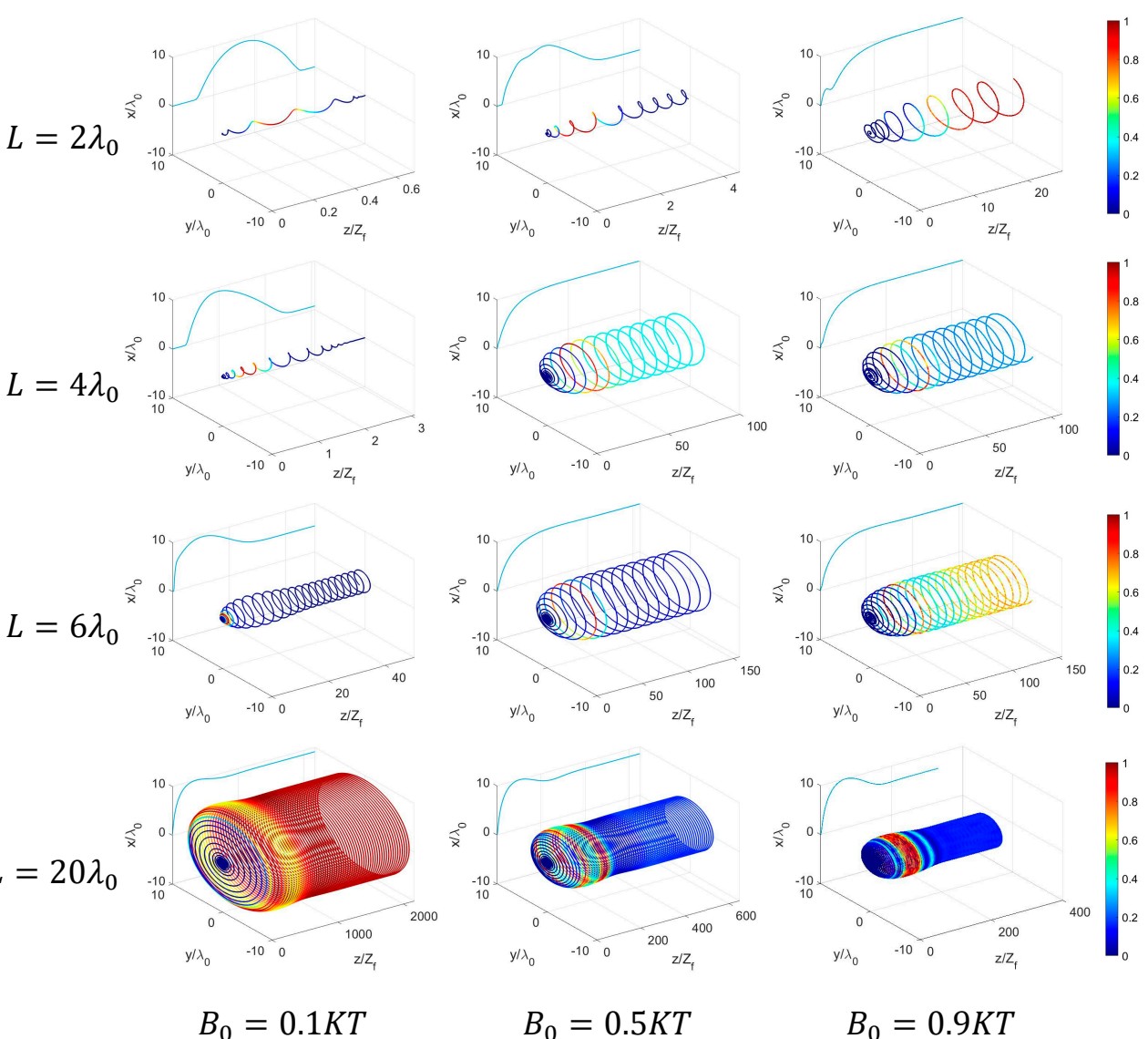

**Figure 3.** Trajectories of electrons and coincidence diagrams normalized to their own maximum radiated power.

First, we can see that several plots in the upper left corner of Figure 3 have a different trajectory compared to the rest of the plots. Their trajectory takes on the shape of an olive, while the trajectory of the other plots has a near-cylindrical shape.

This is because when the pulse length is small and the magnetic field strength is low, the contact time between the electrons and the laser pulse is very short. This also leads to the situation where, in disengaging from the laser field, the magnetic field does not have sufficient effect on the electrons to maintain large energy in the final motion. Ultimately, the electrons move in a nearly straight line along the *z*-axis.

This also leads to the fact that when the electrons disengage from the laser field, the magnetic field does not have a large enough effect on the electrons to give them a large amount of energy in their final spiral motion. Therefore, the electrons end up moving in an almost-straight line along the *z*-axis.

As the magnetic field increases, more energy can be conserved by the magnetic field, and the eventual spiral motion becomes more pronounced as a result. As the pulse length increases, both the rising and falling edges of the laser field become flat, thus lengthening the contact time of the electrons with the laser field. In this way, the duration of action of the magnetic field increases so that ultimately, more energy can be conserved, leading to a helical motion.

By observing the opposition of radiation, we notice that the maximum radiated power occurs near the end of the rising edge of the laser pulse. According to Equations (9) and (10), we know that the radiated power of an electron is proportional to the eighth power of the relativistic factor of the electron $\gamma^8$ and also proportional to the square of the magnitude of the acceleration in the x-direction $d_t u_x^2$. Therefore, we can calculate that the maximum radiated power of the electron occurs at the rising edge of the laser pulse, where $\gamma^8 d_t u_x^2$ is maximized.

This occurs because for a laser field of a given laser intensity, beam waist radius, and magnetic field strength, the time of contact of the laser with the electrons becomes longer as the pulse length increases. As a result, the energy acquired by the electrons in the rising-edge portion of the laser field increases until the electrons are in complete contact with the laser field. After that, the maximum velocity of the electrons remains essentially constant as the pulse length increases.

The situation is different when the pulse length is too large; i.e., pulse length $L > 6\lambda_0$. The excessive pulse length results in a reduced gradient between the rising and falling edges of the laser field, which means that the acceleration of the electrons is diminished. The maximum velocity is the same since the peak amplitude $a_0$ of the laser field is constant and the energy given to the electrons by the laser field is also constant; however, due to the decrease in the gradient, this leads to a decrease in $\gamma^8 d_t u_x^2$ such that the maximum radiated power of the electrons decreases instead as the pulse length increasing.

The contact time of the laser field with the electrons is shorter at pulse lengths $L = 2\lambda_0$. Therefore, the electrons undergo an almost-unobservable falling edge phase after the rising edge of the laser field; i.e., they fail to exhibit the reduced radiated power that they should have exhibited. This is manifested by the smooth motion of the electrons after contact with the laser field, and the radiated power of the electrons remains red after they are out of contact with the laser field; i.e., this not much different from their own maximum radiated power.

At pulse lengths $L = 4\lambda_0, 6\lambda_0$, the contact time of the laser field with the electrons becomes longer. As a result, the deceleration process, which was originally unobservable, i.e., the part of the radiated power that becomes smaller, also becomes visible. When the magnetic field is small, the power radiated by the electrons in the back half is still not far from the maximum power radiated. The reason for the appearance of this phenomenon will be explained in Section 3.3.

At pulse lengths $L = 20\lambda_0$, we find a new phenomenon: a clear phenomenon of double peaks in the image; i.e., the radiated power decreases and then rises, and the rise is located at the position where the radial radius of the electron trajectory is transformed from decreasing to smooth. It is worth noting that the relative magnitude of the radiated power of the electrons in the final spiral part and at the maximum radiation varies for different magnetic field strengths. The radiated power of the electrons is very close when

the magnetic field is small and differs by one or two orders of magnitude when the field is large. The reasons for this phenomenon will also be explained later.

### 3.3. Electron Motion and Corresponding Radiation Properties 3

The parameters of Figure 4 are consistent with those above, with the difference being that the radiated power in Figure 4 is normalized by the maximum value of the radiated power in each set of corresponding pulse lengths L. Figure 4 shows that the magnitude of radiated power in the same set of electron radiation varies with the pulse length L.

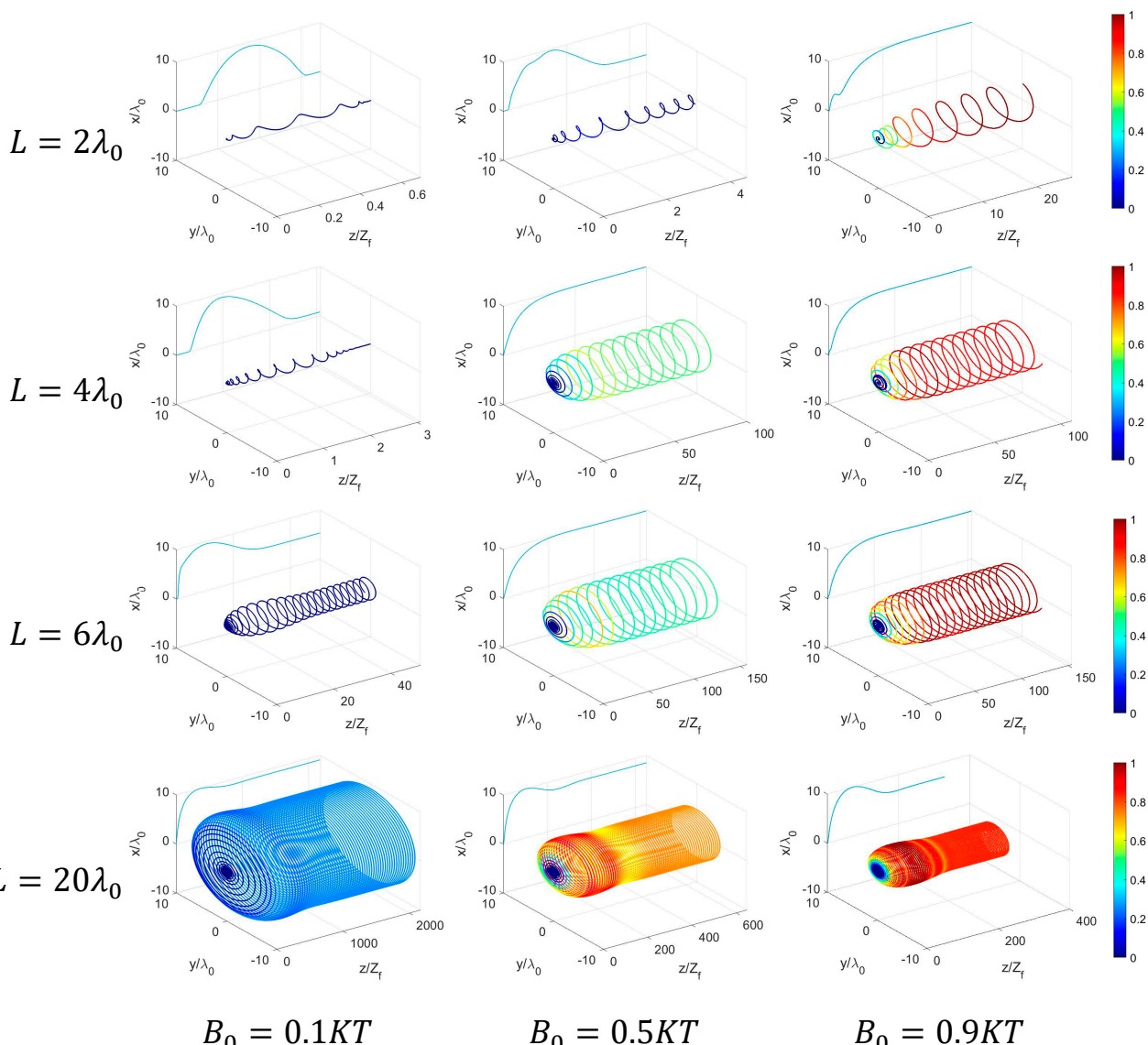

**Figure 4.** Recombination plot of electron trajectories and maximum radiated power normalized to the same pulse length.

By varying the parameters of the normalisation in this way, it is possible to study the causes of the generation of radiation more successfully.

Since the maximum radiated power produced by the same set of pulse lengths L differs by a maximum of about 7 orders of magnitude, in order to observe and represent the variation of the radiated power magnitude with the pulse length L more clearly, we apply some linear transformations to the normalization.

The specific formula is as follows:

$$P_{fix} = \begin{cases} 1 & , P = P_{max} \\ 1 - \frac{log_{10}\left(\frac{P_{max}}{P}\right)}{c} & , \frac{P_{max}}{P} > 10^c \\ 0 & , \frac{P_{max}}{P} \leq 10^c. \end{cases} \tag{12}$$

$P_{fix}$ is the value corresponding to the linear transformation and normalized; and c is the transformation parameter, which, in Figure 4, is c = 7.

It can be seen from the figure that for the same set of pulse lengths L, there is a significant increase in the radiated power of the electrons as the magnetic field strength $B_0$ increases. Moreover, the distribution of the radiated power of the electrons varies as the pulse length L increases. When the pulse length L is large, the ratio of the radiated power of the electrons in the smaller magnetic field strength to the maximum radiated power of the whole group also increases significantly.

By looking at the $L = 4\lambda_0, 6\lambda_0$ sets in Figures 3 and 4, we find that the power radiated by the electrons at the front end of the electron motion and at the back end of the helix increases as the magnetic field strength $B_0$ increases. However, as can be seen from the figure, the radiated power at the back end of the spiral where the electrons are moving increases to a significantly lesser extent than that at the front end of the movement.

This is because at the back end of the helix, the electrons are moving; the laser field is already out of contact with the electrons, and at this point, the only thing controlling the trajectory of the electrons is the magnetic field. Since the magnetic field is nonuniformly strong, the electrons still exchange energy as they move. Thus, the greater the strength of the applied magnetic field, the greater the radiated power emitted from the back end of the helix where the electrons are moving; while at the maximum radiation power at the rising edge of the laser field, the electrons are subjected to the combined action of the laser and magnetic fields. Since the magnitude of the radiated power of the electron is related to the velocity and acceleration of the electron, and the applied nonuniform magnetic field has a periodic acceleration and deceleration effect on the electron, both the velocity and acceleration at the maximum radiated power increase with the applied magnetic field, and the increase is greater than that in the part of the helix at the back end where only the magnetic field is applied.

Comparing the $L = 20\lambda_0$ groups of Figures 3 and 4, we find the phenomenon of double peaks.

This is because there is a process of acceleration to deceleration with respect to the electron's velocity towards the end of the rising edge of the laser field. At this point, there is a brief moment when the acceleration is equal to 0 and the magnitude of the radiated power is 0. As a result, there is a process in which the radiated power decreases from a larger position to zero and increases again to a larger value—the so-called bimodal phenomenon. And the reason why the electrons only show double peaks when the pulse length L is large is because when the pulse length is large, the gradient between the rising and falling edges is smaller, and the acceleration and deceleration are less pronounced. As a result, the radiated power changes more slowly, which is manifested in the trajectory of motion, which is more pronounced.

### 3.4. Electron Motion and Corresponding Radiation Properties 4

The parameters of Figure 5 are consistent with those above, with the difference being that the radiated power of Figure 5 is normalized by the maximum value of the radiated power in each set of corresponding magnetic field strengths $B_0$. The normalization is conducted via a linear transformation, consistent with the above, corresponding to the parameter C = 7.

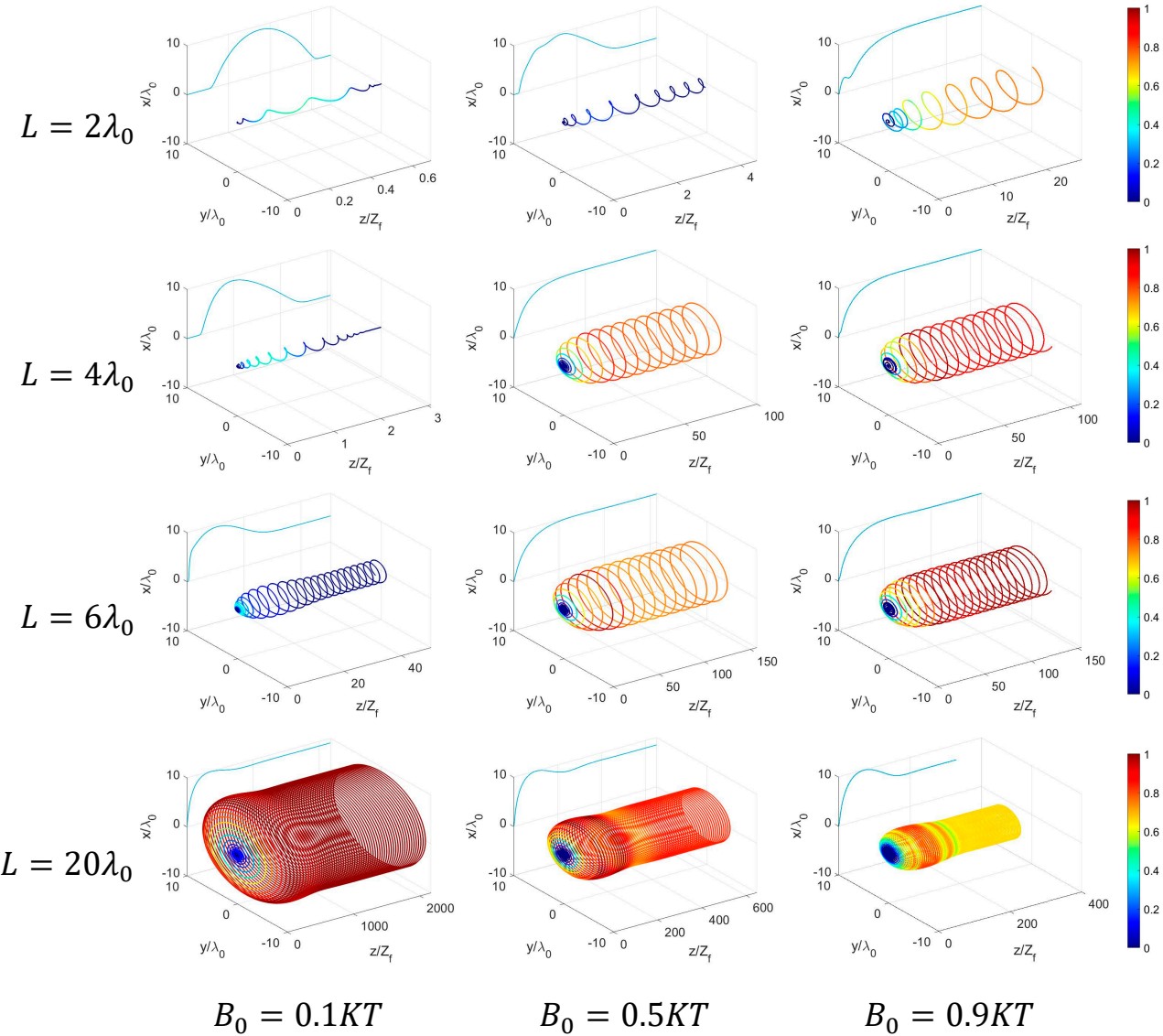

**Figure 5.** Recombination diagram of the electron's trajectory and the maximum radiated power normalized to the same magnetic field.

Looking at Figure 5, we can see that as the magnetic field strength increases, the radiated power emitted by electrons with smaller pulse lengths differs from that emitted by electrons with larger pulse lengths by several orders of magnitude to a small difference.

This is because when the pulse length and magnetic field strength is small, the contact time between the laser field and the electron is short, not providing sufficient energy, so the magnetic field can save less energy. When the contact time between the laser field and the electrons increases, the magnetic field is able to save the energy; the energy is therefore larger, so the difference in the radiated energy of the electrons at this time is not large.

We also find that when both the magnetic field and the pulse length are exceptionally large, a situation occurs wherein the radiated power of the final spiral cylindrical portion of the electron is much less than the maximum radiated power due to the fact that with a larger pulse length, the electron consumes more time and is subjected to more pronounced deceleration when it is removed from the action of the laser field, which is significantly increased by the presence of the magnetic field, so that at times when both the pulse length and the magnetic field are both larger, this occurs.

Comparing the plots in group (a) of Figures 4 and 5, we can see that when the pulse length is small, the color at the maximum radiated power in Figure 4 is dark blue, while

the color at the maximum radiated power in Figure 5 is light blue. This indicates that the increase in magnetic field has a more significant increase in radiated power.

### 3.5. Space Radiation Properties of Electrons

The parameter group settings of Figure 6 are the same as above, where its radiated power magnitude is normalized by the maximum value of every parameter group itself. The maximum radiated power and the corresponding maximum radiated polar angle for each specific parameter group will be given in the next section.

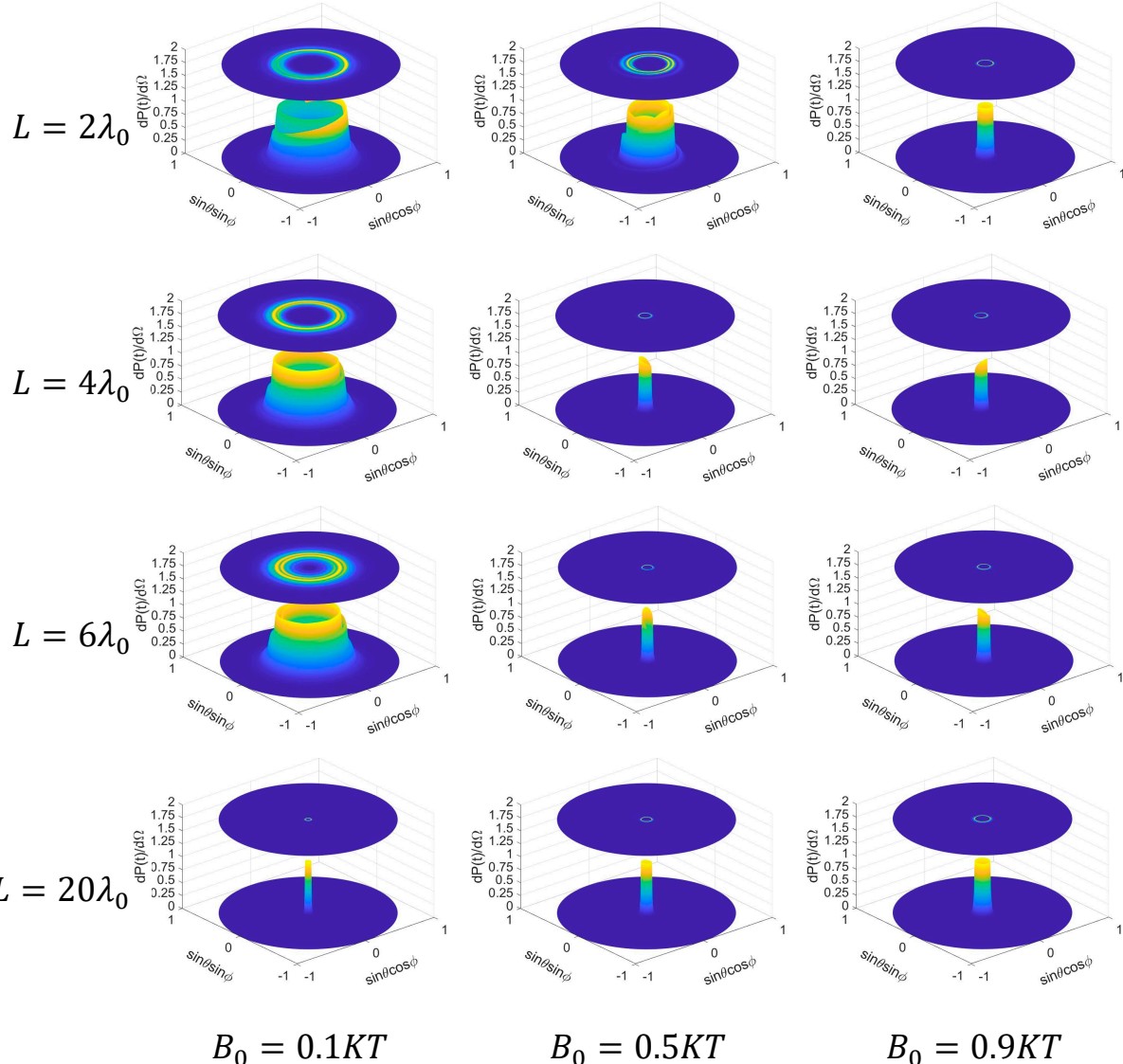

**Figure 6.** Space radiation properties of electrons.

The above figure is drawn according to the spherical coordinate system of the electron spatial radiation, where the horizontal and vertical axes are sinθsinφ and sinθcosφ, respectively, where $\varphi$ is the azimuthal angle and $\theta$ is the polar angle, as shown in Figure 1. The corresponding $z$-axis coordinate value represents the radiation power magnitude $dP(t)/d\Omega$. From the figure, it is evident that as the strength of the magnetic field increases, the radiated power of the electrons of different solid angles tends to be the same, and phenomena involving significant magnitude changes in Figure 6 ($L = 2\lambda_0$; $B_0 = 0.1KT$) become much less pronounced. The corresponding collimation improves, indicating that the radiation is more concentrated.

It is worth noting that when the pulse length is too large, i.e., $L = 20\lambda_0$, the collimation of the electron's radiation deteriorates slightly with the growth of the magnetic field strength $B_0$. This is because the spatial radiation of electrons is more closely related to the trajectory of motion. With the increase in the magnetic field strength $B_0$, the electron motion trajectory changes from a spiral with a constant radial radius to a spiral whose radial radius first decreases and then stays constant. As shown in the previous section, a decreasing radial radius corresponds to a larger radiation emission. Moreover, since the direction of the radiation is the same as that of the motion of electrons, the collimation of radiation corresponds to poorer collimation.

### 3.6. Radiated Power of Electrons versus Maximum Radiated Polar Angle

Figure 7 shows the variation in the maximum radiated power $dP(t)/d\Omega$ and the maximum radiated polar angle $\theta_r$ with magnetic field strength at different pulse lengths. The pulse lengths $L = 2\lambda_0, 4\lambda_0, 6\lambda_0, 20\lambda_0$, respectively, in Figure 7.

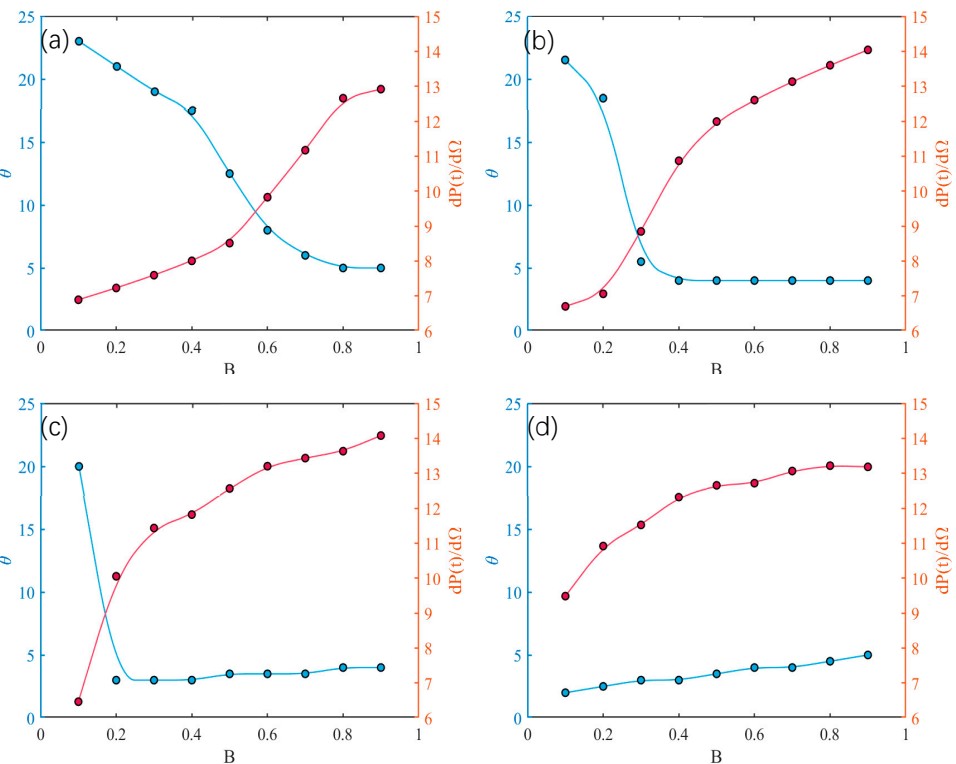

**Figure 7.** Maximum radiated power of an electron and the relation between the polar angle of maximum radiated power and magnetic field strength. Where (**a**–**d**) represents the pulse lengths $L = 2\lambda_0, 4\lambda_0, 6\lambda_0, 20\lambda_0$.

It can be seen from the figure that the maximum radiation polar angle is approximately the same in every plot when the magnetic field intensity $B_0 = 0.9$. This is because collimation is better with a smaller magnetic field intensity at a larger pulse length. This is because a larger pulse length corresponds to a smaller gradient along the rise. In this way, electrons are affected by the laser pulse for a longer period of time when they reach the focal point of the laser field, leading to greater velocity in the Z direction when the maximum radiation is emitted, as well as a smaller maximum radiation polar angle.

When the pulse length is too big, the collimation of radiation decreases as the magnetic field intensity $B_0$ increases. This is because with the increase in $B_0$, the radial radius of the electron also increases, and the acceleration of the electron's circular motion increases. Thus, the velocity in the $z$-direction is weakened when the maximum radiation is emitted, which also leads to a decrease in the polar angle of the maximum radiation.

## 4. Conclusions

In summary, we have investigated the electron motion properties and space radiation properties resulting from nonlinear Thomson scattering of stationary single electrons by a tightly focused circularly polarized laser in the presence of an applied magnetic field. Creatively, we analyze it from the point of view of the radiation emitted during the motion. The trajectory and radiated power variations under different magnetic field values and different pulse length values are investigated. And the effects of pulse length and magnetic field on the collimation of radiation of electrons are analyzed.

From the perspective of the radiated power of the electrons, the maximum radiated power of the electrons increases with the pulse length L when the magnetic field strength $B_0 \leq 0.3$. When the magnetic field intensity is greater than 0.3, the maximum radiated power of the electrons occurs at $L = 6\lambda_0$, after which it decreases slowly.

Analyzing from the point of view of the collimation of electrons, the collimation of the radiation of the electrons consistently improves with the increase in pulse width. When the pulse length $L$ is small, collimation increases with the increase in magnetic field; while when the pulse length $L > 6\lambda_0$, collimation slowly worsens with the increase in the magnetic field.

Therefore, under the premise that the radius of the laser pulse waist and the peak amplitude of the electrons are certain, obtaining high-energy and highly collimated X-rays can be realized by increasing the pulse length up to $6\lambda_0$ and increasing the magnitude of the applied magnetic field. We can also achieve this purpose by continuously increasing the pulse length and applying a smaller magnetic field. The choice between the above two options can be flexible and based on different requirements.

This provides a good modulation scheme for the use of nonlinear Thomson scattering in scientific experiments, as well as some potential applications.

**Author Contributions:** Conceptualization, Y.Z., Y.W., Q.Y. and Y.T.; Methodology, H.W., F.G., Y.Z. and Y.W.; Software, H.W., Y.Z., Y.W., Q.Y. and Y.T.; Validation, H.W. and F.G.; Writing—original draft, H.W.; Writing—review & editing, H.W.; Visualization, H.W.; Project administration, Y.T.; Funding acquisition, Y.T. All authors have read and agreed to the published version of the manuscript.

**Funding:** This work has been supported by the National Natural Sciences Foundation of China under grants No. 10947170/A05 and No. 11104291; the Natural Science Fund for Colleges and Universities in Jiangsu Province under grant No. 10KJB140006; the Natural Sciences Foundation of Shanghai under grant No. 11ZR1441300; the Colleges and Universities in Jiangsu Province under grant No. 10KJB140006; and the Natural Science Foundation of Nanjing University of Posts and Telecommunications under grant No. 202310293146Y.

**Data Availability Statement:** The raw data supporting the conclusions of this article will be made available by the authors on request.

**Conflicts of Interest:** The authors declare no conflicts of interest.

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
