# Peer review of "Analysis of the Effect of Pulse Length and Magnetic Field Strength on Nonlinear Thomson Scattering"

_applsci, doi:10.3390/app14156776_

Round 1

Reviewer 1 Report (Previous Reviewer 1)

Comments and Suggestions for Authors

Authors did editing which improved the manuscript

Author Response

thank you for your review

Reviewer 2 Report (Previous Reviewer 3)

Comments and Suggestions for Authors

In the revised version of their manuscript, the authors corrected some formulas as well as the value of the laser frequency (from 1 nanometer to 1 micrometer).

My remaining comments/suggestions were ignored.

There is no point-by-point response to my remarks. In view of this, I recommend that the paper be rejected.

Comments on the Quality of English Language

The paper is often hard to read. I suspect that in general the authors mean pulse length when they write pulse width..

Author Response

Reviewer 3 Report (New Reviewer)

Comments and Suggestions for Authors

I have read through the manuscript by Wang et al., where they discussed the influences of the laser width and magnetic field strength applied externally on the electron trajectories and radiation. These results may be of interest for experimentalist, and this can play a guiding reference for readers. Before I can make a final recommendation, several minor issues should be addressed:

1. Several descriptions are not rigorous. For example, what is the amplitude of electrons as mentioned in the Abstract,

2. Fig2-6 are not clear to see the coordinates. Please provide a high resolution figures int he revised version.

3. Why you add this strange sentences here in line 202. I noticed you have several co-authors.

In line 230, the last sentence is not complete. Please amend it.

In line 264, what are D groups?

4. Why did you use 4 subsections called “electron motion and corresponding radiation properties”. This looks strange.

5. Line 331, it should be figure 7, not figure 6.

Comments on the Quality of English Language

It should polished by a native english speaker.

Round 2

Reviewer 2 Report (Previous Reviewer 3)

Comments and Suggestions for Authors

When I was asking to specify the laser intensity I meant, of course, units that are generally used, such as W/cm^2, not the dimensionless units used by the authors (a0 = 5). I still did not find the initial electron velocity mentioned.

I did not say that references to papers that are available only in Chinese are unsuitable. I suggested to replace them by equivalent ones IF POSSIBLE. If not, the Chinese refs should stay.

Comments on the Quality of English Language

There is still room for improvement, but understanding the message of the paper is not inhibited.

Author Response

When I was asking to specify the laser intensity I meant, of course, units that are generally used, such as W/cm^2, not the dimensionless units used by the authors (a0 = 5). I still did not find the initial electron velocity mentioned.

I apologise for my oversight. I have added this part of the description to the abstract.

I did not say that references to papers that are available only in Chinese are unsuitable. I suggested to replace them by equivalent ones IF POSSIBLE. If not, the Chinese refs should stay.

I agree with you so much that I've changed the correspondence in the article

This manuscript is a resubmission of an earlier submission. The following is a list of the peer review reports and author responses from that submission.

Round 1

Reviewer 1 Report

Comments and Suggestions for Authors

Analysis of the effect of pulse width and magnetic field 2 strength on nonlinear Thomson scattering

The study investigates electron motion and space radiation resulting from nonlinear Thomson scattering when stationary single electrons interact with a tightly focused circularly polarized laser in an applied magnetic field. The authors examine the emitted radiation and explore electron trajectories, radiated power changes, and the effects of magnetic field and pulse width variations on electron collimation. The findings reveal that maximum radiated power increases with pulse width until a magnetic field threshold, beyond which it gradually decreases. Moreover, wider pulse widths consistently improve electron collimation, although this effect diminishes with increasing magnetic field strength. 

I think this article should not  published in applied science journal, my concerns are the following:

1.     I don’t find much novelty in the work

2.     Hard to follow the paper

3.     Some experimentation to support the theory would be better

4.     Figure description is not adequate, the title of the Figures should improve

Comments on the Quality of English Language

Some sentences are hard to understand. Figures 4-6 titles are not fully visible. Titles of Figures can be improved by describing sub-figures.

Author Response

Thank you for pointing this out. We agree with this comment. We have improved our cover letter to include detailed descriptions of innovative content in our manuscript, hoping it will be helpful to you.

Reviewer 2 Report

Comments and Suggestions for Authors

The paper is well-written, with a clear and organized presentation of the research. I particularly appreciate the effort invested in outlining the methodology and presenting the results coherently. I only have a few suggestions/questions.

Could you change or add keywords to better describe the specific focus and findings of your paper?

Considering the potential applications of nonlinear Thomson scattering in generating high-energy X-rays, are there any specific technological implications of the research findings? For example, could the insights gained from this study contribute to the development of more efficient X-ray generation techniques?

Author Response

 Thank you for pointing this out. We agree with this comment.We have updated our keywords to include ‘external magnetic field,’ and revised our cover letter to highlight how our research contributes to advancements in engineering

Reviewer 3 Report

Comments and Suggestions for Authors

This paper is almost impossible to read. 

It takes a while to figure out that the authors consider a copropagating geometry. There is hardly any discussion of the background of the paper.

The parameter values underlying the study are hard to find and they are strange: an initial laser-field wavelength of 1 nm (corresponding to a frequency of about 1 keV), the laser intensity is only given in units of mc^2/e and corresponds to an unrealistically strong field for the frequency of 1 keV, and I could not find a value of the initial electron energy. The frequency spectrum of the emitted radiation I found nowhere mentioned.

In Eq. (2) it should be rho^2/b^2 in the exponent or else an approximation has been introduced. The ellipticity delta is not introduced. For the simulations a circularly pulse is subsequently employed. I would strongly discourage the nomenclature "al" introduced in Eq. (2). The reader will interpret this as a product of a and l and look in vain for the definition of a and l. In Eqs. (7), the quantity "del" appears, which is probably the ellipticity. Also in Eq. (7), the notation "+ ..." could be misinterpreted as meaning there are additional terms that have been omitted. The figures are taken straight from the MATLAB output; the axis labels are unreadable and in some cases the figure overlaps the caption making it unreadable. The authors are using "pulse width" for what I suppose is "pulse length" (length refers to the longitudinal extent of the pulse, width to the transverse). Citing papers that are written in Chinese (Refs. 15 and 16) will not be helpful to many readers; aren't there additional and equivalent references in English?

In the discussion of the results, the English is complicated and wordy and very hard to follow.

I cannot recommend publication of the current version of the paper. A thoroughly revised version may be publishable provided the parameters used are at least borderline realistic.

Comments on the Quality of English Language

See above; maybe the authors can have the English checked by a qualified person.

Author Response

 Thank you for pointing this out. We agree with this comment.I am sorry for our mistakes.We have made modifications to your suggestions in the manuscript and highlighted them. It’s worth noting that the wavelength of our laser field is 1 micron. We sincerely apologize for our mistake.